# Spatial scale of receptive fields in the visual sector of the cat thalamic reticular nucleus

Cristina Soto-Sánchez[1,2], Xin Wang [1,3], Vishal Vaingankar[1], Friedrich T. Sommer[4] & Judith A. Hirsch[1]

Inhibitory projections from the visual sector of the thalamic reticular nucleus to the lateral geniculate nucleus complete the earliest feedback loop in the mammalian visual pathway and regulate the flow of information from retina to cortex. There are two competing hypotheses about the function of the thalamic reticular nucleus. One regards the structure as a thermostat that uniformly regulates thalamic activity through negative feedback. Alternatively, the searchlight hypothesis argues for a role in focal attentional modulation through positive feedback, consistent with observations that behavioral state influences reticular activity. Here, we address the question of whether cells in the reticular nucleus have receptive fields small enough to provide localized feedback by devising methods to quantify the size of these fields across visual space. Our results show that reticular neurons in the cat operate over discrete spatial scales, at once supporting the searchlight hypothesis and a role in feature selective sensory processing.

[1] Department of Biological Sciences and Neuroscience Graduate Program, University of Southern California, 503 HNB, MC 2520, 3641 Watt Way, Los Angeles, CA 90089-2520, USA. [2] Biomedical Research Networking Center in Bioengineering, Biomaterials and Nanomedicine (CIBER-BBN), Bioengineering Institute, Miguel Hernández University (UMH), Avda. Universidad s/n, 03202 Elche, Spain. [3] Computational Neurobiology Laboratory, The Salk Institute for Biological Studies, 10010 North Torrey Pines Road, La Jolla, CA 92037, USA. [4] Redwood Center for Theoretical Neuroscience-HWNI, University of California at Berkeley, 575A Evans Hall, MC 3198, Berkeley, CA 94720-3198, USA. Cristina Soto-Sánchez and Xin Wang contributed equally to this work. Correspondence and requests for materials should be addressed to J.A.H. (email: jhirsch@usc.edu)

The thalamic reticular nucleus (TRN) is a thin network of inhibitory cells that has been likened to a "guardian of the gateway"[1] because of its position with respect to thalamus and cortex and its connections with these structures. The visual

sector of the TRN, called the perigeniculate nucleus (PGN), lies above the lateral geniculate nucleus (LGN) (Fig. 1a–c). The PGN receives descending input from the visual cortex and ascending input from relay cells[2, 3], which it inhibits in return[4] to form the

**Fig. 1** Neural circuits and sample recordings from the visual thalamus (PGN and LGN) of the cat. **a** The PGN and main layers of the LGN drawn as *rectangles* including populations of excitatory (*light fill*) and inhibitory neurons (*dark fill*). Glutamatergic synapses (*open arrows*); GABAergic synapses (*filled circles*) and electrical synapses (resistor). **b** Parasagittal section of the visual thalamus; the layers of the LGN (*gray*) and the PGN (*blue*). *Red lines* and corresponding text mark retinotopic position[7]; *dashed box* illustrates a single retinotopic location sampled by the multi-electrode. **c** Nissl-stained coronal section showing a recording site (*red arrow*). **d** Recordings from a neuron in the PGN (*blue*) and a relay cell in the LGN (*black*) with bursts (*red*) shown at an expanded time scale. Illustrations of the (**e**) thermostat and (**f**) searchlight hypotheses. Boxes with *dashed outlines* represent local circuits for adjacent regions of the visual field; excitatory, inhibitory, and electrical synapses as above. For the thermostat hypothesis (**e**), reticular feedback has a net inhibitory affect, homogenizing thalamic activity (*bottom bar graph*) by suppressing peaks of activity driven by feedforward input. Because single reticular neurons must sense distributed thalamic activity, their receptive fields (*top*) are larger than those of a thalamic relay cell (*bottom*). For the searchlight hypothesis (**f**), reticular feedback exerts a net positive effect, enhancing local activity in the thalamus (*bottom bar graph*) through rebound excitation. Because reticular neurons signal top-down attentional spotlights, their receptive fields sizes (*top*) are comparable to those in the LGN (*bottom*)

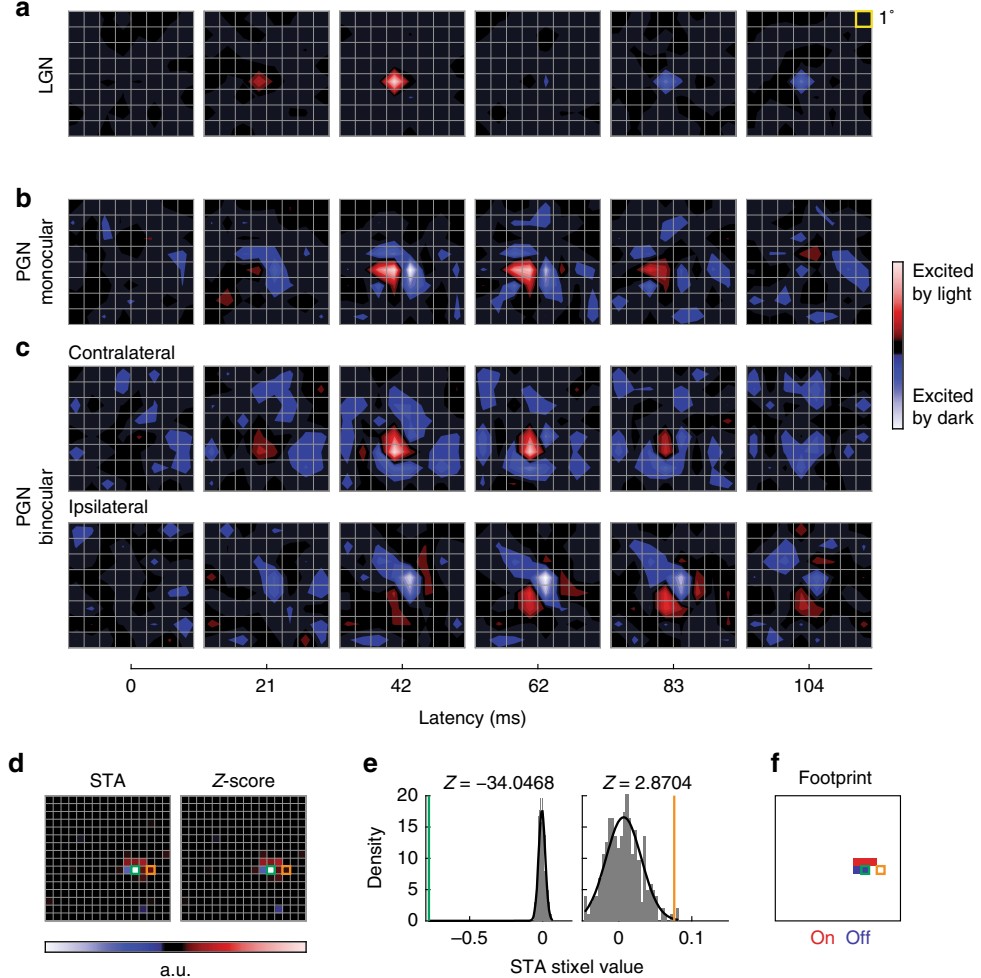

**Fig. 2** Quantifying spatiotemporal receptive fields in the LGN and PGN. **a–c** Spatio–temporal receptive fields estimated by reverse correlation of spikes to Gaussian white noise shown as a series of snapshots (two-dimensional spatial maps) obtained at different latencies for an example cell in the LGN (**a**), a monocular cell in the PGN (**b**), and a binocular cell in the PGN (**c**). **d–f** Quantification of the spatial scale of each map (STA) by estimating its "footprint". $Z$-scores were estimated for each stixel in the STA by bootstrap resampling. **d** Snapshot of an example STA and its corresponding $Z$-score map. **e** $Z$-scores for two example stixels (outlined in *green* and *orange* in **d**) computed with the bootstrap. *Color-coded bars* indicate the actual STA value (effectively, stimulus contrast) of the stixel, the *gray histograms* plot the bootstrapped STA values for each stixel, and the *black curves* show the Gaussian fit of the distribution. The $Z$-scores were used for subsequent Benjamini–Hochberg multiple-hypothesis testing to determine significant regions of each STA. **f** Time-collapsed spatial map of all significant stixels form the "footprint" used to quantify the spatial scale of the receptive field

first feedback loop in the early visual pathway[5, 6] (Fig. 1a). This inhibitory feedback is topographically organized[5, 7, 8] (Fig. 1b), a pattern shared by circuits in the auditory[9] and somatosensory[10, 11] sectors of the TRN, and their target thalamic nuclei. In addition, response properties of neurons in the LGN and PGN are different from one another, not only in terms of firing patterns[12, 13], but also visual response. Unlike relay cells and interneurons in the LGN, which are almost always monocular and whose receptive fields resemble those in retina[14–17], neurons in the PGN are often binocular[5, 18–21] and selective for complex visual features, reminiscent of cortex[21]. Thus, the PGN contains a retinotopic representation of the visual world that is in register with, but different from, that in the LGN.

What role might the PGN or other sensory sectors of the TRN play in information processing? Two leading hypotheses provide a starting point for addressing this question. In the first scenario, the PGN senses distributed activity levels in the LGN, exerting negative feedback; to quote Crick, "the function of the reticular complex would be to act as an overall thermostat of thalamic activity"[1]; we refer to this as the thermostat hypothesis (Fig. 1e).

The competing view, the searchlight hypothesis, holds that PGN mediates top-down attentional signals "so that "attention" will be focused on the most active thalamo-cortical regions"[1] (Fig. 1f).

The searchlight hypothesis predicts three independent mechanisms. First, the top-down signals that encode spatial attention should modulate activity in the PGN. Second, this descending input should increase the activity of relay cells whose receptive fields fall within the area of interest. Third, reticular neurons should have localized visual responses able to exert a focal influence over the region of interest. The third prediction, which is the focus of the current study, is equally important to sensory processing per se as it suggests that reticular input has a role in processing spatially discrete visual features.

There has been strong experimental evidence in support of the first two mechanisms. For example, when an animal attends to a target within the neural receptive field, firing rates are reduced in TRN and increased in LGN[22, 23], optogenetic modulation of the TRN regulates attentional state,[23–26] and genetic impairment of reticular output leads to attentional deficits[24–26]. However, the spatial scale at which neurons in the PGN operate remains

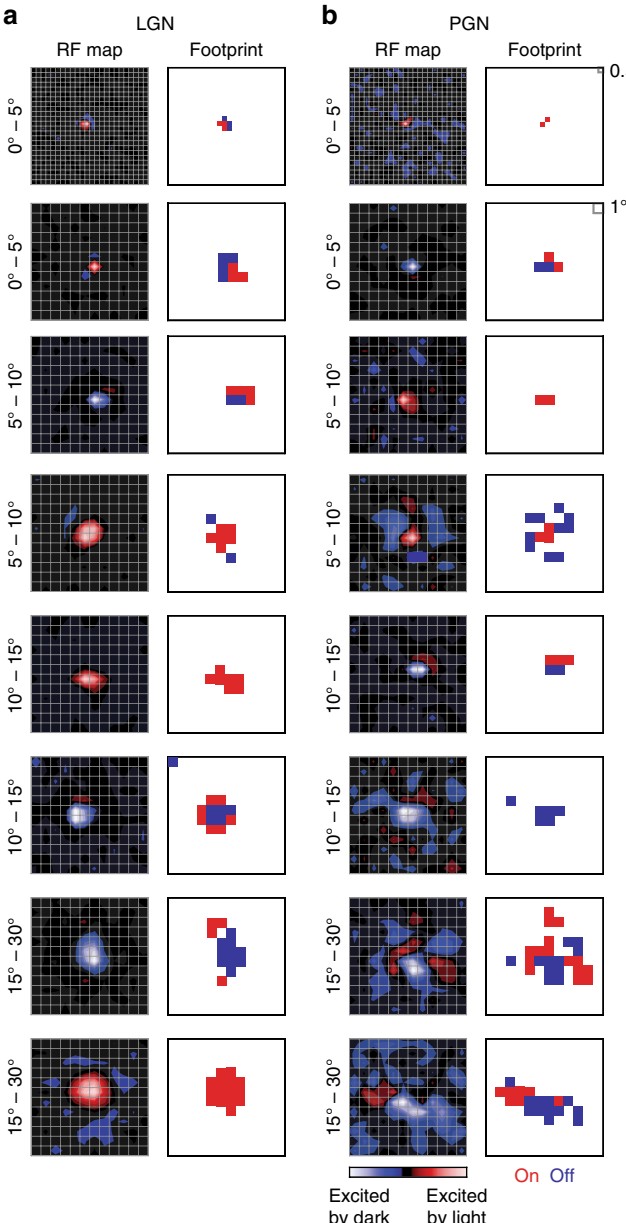

**Fig. 3** Spatial scales of receptive fields in the LGN and PGN are comparable. **a** Spatial receptive fields (*left column*) and footprints (*right column*) of eight example LGN relay cells ranked by their eccentricities (as labeled). **b** Examples of PGN receptive fields and footprints at matching eccentricities

unclear. Receptive fields there have diverse shapes that cannot be quantified using standard methods. Thus, at present, there are only qualitative, and sometimes contradictory, accounts[20, 27, 28] about receptive sizes of reticular neurons.

Hence, we devised a method to map and quantify the spatial scale of receptive fields of neurons in the PGN and compare these results to a complementary data set from the LGN. Specifically, we adapted statistical approaches used in functional magnetic resonance imaging (fMRI)[29] for the analysis of neural responses to white-noise stimuli. Just as fMRI reveals brain regions whose activity correlates with specific behaviors, our method reveals regions of visual space, that we call footprints, from which statistically significant neural responses are evoked. Our results show that the sizes of footprints in PGN and LGN are quantitatively comparable and scale similarly with eccentricity.

This suggests that reticular circuits are able to operate at local spatial scales.

## Results

**Quantifying the sizes of diversely shaped receptive fields.** Our sample includes 88 neurons in PGN and 89 in LGN from 23 adult cats. Most receptive fields (95%) fell within a 15° radius from the area centralis, with the rest reaching as far as 30° away. The fields were mapped using standard methods of reverse correlation and the resulting spike-triggered averages (STAs) of the stimulus ensemble are displayed as a sequence of contour plots that show the temporal evolution of the response (Fig. 2a–c); the stimulus was a sequence of checkerboards in which the luminance of each component pixel was independently drawn from a Gaussian distribution. All cells in the LGN were monocular and had circular on-center or off-center receptive fields, with an example depicted in Fig. 2a (note, the stimulus we used lacks the spatial coherence necessary to drive strong responses from the surround). Unlike the situation in the LGN, receptive fields in the PGN varied widely in shape and complexity (e.g., Fig. 2b, c) and ~26% of reticular cells in our data set were strongly binocular (e.g., Fig. 2c). Additional studies with sparse-noise stimuli showed that ~65% of cells in the PGN responded well to overlapping bright and dark stimuli (data not shown) and see ref. [21].

Our first task was to characterize the spatial extent of receptive fields in the PGN in a manner that permitted direct comparison with those in the LGN at equivalent eccentricities in visual space. The diverse and even patchy shapes of reticular receptive fields (e.g., Fig. 2b, c) precluded the use of standardized fitting functions like the Difference of Gaussians frequently employed to describe receptive fields in the retina or the LGN. Thus, we developed a new, non-parametric method that combined three approaches (see Online Methods): standard reverse-correlation analysis, bootstrap resampling by randomly shifting spike trains[30], and statistical methods used in fMRI to localize task-related changes in brain activity[29]. In essence, we used our method to identify pixel locations that evoked statistically significant responses, as explained in Fig. 2d, e. We called the resulting maps "footprints". Because our technique allowed us to map receptive fields of any shape, we were able to analyze receptive fields in the LGN and PGN in the same way and compare them across the visual field. Example receptive fields and corresponding footprints of cells in the LGN and PGN across different eccentricities are shown in Fig. 3a, b. The typical footprints of reticular neurons were only slightly larger than those of relay cells at similar eccentricities. Very large reticular receptive fields were rare and most common in peripheral regions of visual space. Last, the pattern of the footprints in LGN differed from those in PGN, reflecting the difference between receptive fields with a center surround vs. patchy structure (and see below). This difference is quantified in Supplementary Fig. 1.

By measuring the areas of the footprints, we quantified the spatial scale of receptive fields in the LGN and PGN. Most of the receptive fields we measured lay within 15° of the area centralis (the region that occupies most of the LGN[7], given the magnification factor). Sampling in the far periphery was less dense. We analyzed our data set in several ways. First, we asked if the scale of responses to stimuli of either polarity were similar by comparing the territories covered by bright vs. dark pixels. There was no apparent difference between receptive fields in the PGN vs. the LGN for either on or off footprint sizes (Fig. 4a, *leftmost two columns*). We next compared the territories of the pixels' dominant polarity, on or off, (Fig. 4a, *third column*) for cells in both nuclei. For the LGN, this region corresponded to the receptive field center, but for the PGN this area was defined by

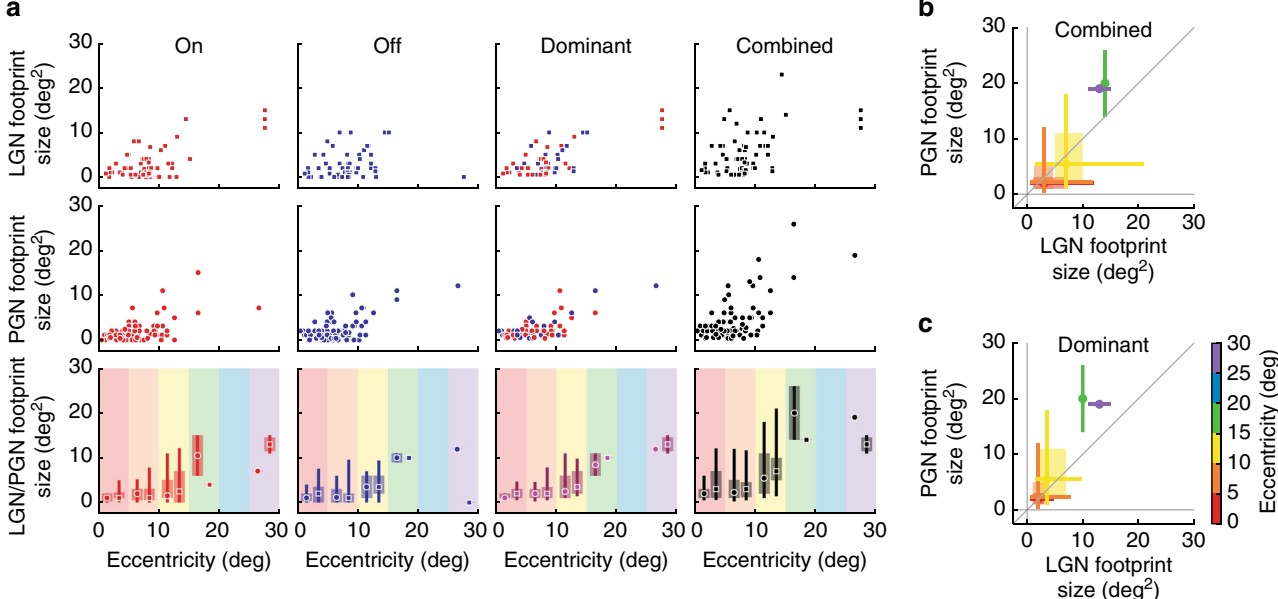

**Fig. 4** Receptive field size as a function of eccentricity. **a** Scatter plots of footprint sizes as a function of eccentricity, for LGN (*square symbols, top row*) and PGN (*circular symbols, middle row*) for either on, off, the dominant, or both stimulus polarities. Note the dominant footprint was on or off, whichever had the larger magnitude, and the combined footprint was the union of the on and off regions. Footprint sizes in the LGN vs. PGN (*bottom row*) increase with increasing eccentricity (binned at 5° intervals); the *square* (LGN) or *circular* (PGN) *symbols* in each bin mark the mean, the *wide pale bars* associated with the symbols denote the quartiles, and the *lines* indicate extreme values. **b, c** A direct comparison of combined (**b**) and dominant (**c**) footprint sizes for PGN (ordinate) vs. LGN (abscissa) across eccentricities; the line of unity slope is in *gray*; conventions as in (**a**). The footprints were calculated with a threshold of $q = 0.01$; see Supplementary Fig. 2 for footprints calculated with more relaxed or more stringent thresholds

the pixels with the same polarity as the pixel with the largest significance value. Finally, we compared the combined, on and off, footprints (Fig. 4a, *rightmost column*). We did not observe significant differences between the two nuclei, with the possible exception of a trend for a relative increase in size of fields in the PGN in the far periphery; this was true whether we compared the entire (i.e., on and off combined) footprint for cells in the LGN and PGN (Fig. 4b) or the dominant footprints (Fig. 4c). Note, the footprints depicted in Fig. 3 and plotted in Fig. 4 were made using a significance criterion of $q = 0.01$ (see "Methods" for details). However, even if the threshold values changed to $q = 0.05$ or $q = 0.001$, receptive field sizes in the LGN and PGN remained comparable, see Supplemental Fig. 2.

Some of the receptive fields in our sample comprised both STAs and additional components (recovered using spike-triggered covariance analysis); in these cases, both types of components were roughly the same size[21]. In addition, almost all reticular neurons that responded to visual stimuli were driven by the Gaussian-noise stimulus we used. Thus, our footprint analysis describes the vast majority of neurons in the visual TRN.

## Discussion

Competing hypotheses about the function of the TRN in visual processing involve different predictions about the relative spatial scales of receptive fields in the PGN and LGN. The searchlight hypothesis assumes that neurons in the TRN are able to exert a focal influence on the LGN, whereas the thermostat hypothesis supposes a widespread effect[1]. Thus, we developed techniques that allowed us to quantify the spatial extent of receptive fields, such as those in the TRN, whose diverse and sometimes scattered arrangements cannot be described using standard parametric methods. We found that neurons in the PGN operate over local spatial scales, on par with relay cells in the LGN. Taken together with the fact that the projection from the PGN to LGN is

topographic[5, 8, 31], our results both corroborate the searchlight hypothesis and support a role for the TRN in bottom-up processing of discrete components of the visual image.

The earliest studies of the PGN suggested that receptive fields there were large, amorphous, and not selective for stimulus contrast[5, 7, 20]. These observations, coupled with the finding that there is a suppressive component of the relay cell's receptive field that extends far beyond the center and surround[14, 32], led to the idea that the PGN provided a distributed, non-selective form of gain control—the thermostat hypothesis.

Accumulating evidence, however, continues to challenge this unilateral view. For example, one key study suggests that the suppressive surround is fed forward from retina[33–35] in primate and, in part, in carnivore[36]. STA and covariance analyses have shown that neurons in the PGN are, in fact, highly selective and are tuned for specific features[21], as are cells in the auditory[31] and somatosensory divisions of the TRN[37]. Further, although many studies of the PGN report large response areas, it is important to recognize that prior investigators used the diffuse expanse of the receptive field as a criterion for identifying the PGN[5, 7, 20]. That said, there have been qualitative descriptions of small receptive fields in the nucleus[21, 22]. Ultimately, our systematic analyses establish that reticular receptive fields are roughly the same size as those of relay cells at a given eccentricity. This finding is in keeping with the searchlight hypothesis and suggests that if inhibitory feedback from the PGN extends far beyond the relay cell's classical receptive field, it is likely mediated by the pooled input of cells tuned to complex features[21].

We had not anticipated our results. Because the PGN is slim and cell sparse, intuition suggests that receptive fields there should be large if they are to represent the entire visual field. Surprisingly, however, only a small number of neurons are required to encode the visual scene; for example, there are fewer than 7000 alpha ganglion cells in each retina of the cat. The neural population of the PGN is likely twice as large[8], also

see Supplementary Material. Further, unlike relay cells, many reticular cells are binocular and on–off, thus potentially reducing the number of neurons required to tile visual space.

A second reason to expect that receptive fields in the PGN would be large is that reticular cells are coupled by gap junctions[38]. However, gap junctions act as low-pass filters and thus convey hyperpolarizing signals most effectively[39]. Also, reticular neurons inhibit their neighbors via GABAergic synapses[40–43]. By way of comparison, cortical interneurons communicate using chemical and electrical synapses[44], and have spatially discrete receptive fields[45].

A remaining concern is that anesthesia might have reduced the size of the receptive fields we measured. This seems unlikely because preference for spatial frequency is either higher or unchanged in the awake vs. anesthetized monkey[46]. Similarly, receptive-field size in visual cortex contracts rather than expands with progressive desynchronization of the electroencephalogram (EEG)[47].

Although, one context for our results involves the spotlight of attention; another perspective involves implications for bottom-up processing. For example, when taken together with earlier work[21], our results show how reticular feedback could influence thalamocortical communication in a feature-specific manner at a fine spatial scale. This influence might enhance selectivity for particular stimulus attributes and/or mediate local gain control. Of course, a fuller test of the roles of the PGN awaits further experiments. It will be important to identify the inputs and outputs of each reticular cell and then determine how functional connectivity changes as a function of stimulus context, behavioral state, and task demands.

Our study involved only the visual sector of the TRN. Although, the spatial scale of receptive fields in auditory or somatosensory TRN has not been measured, reticular cells there are tuned for specific features[37, 48]. These findings suggest that strategies for processing in the visual TRN are conserved across sensory modalities.

Finally, irregular and diversely shaped receptive fields are common, even as early as V1, and thus our new approach to quantifying the size of receptive fields with arbitrary shapes should have broad application. Just as the technique provided insight key to understanding the function of the visual TRN, it may help to understand other stations in the sensory pathway.

## Methods

**Preparation**. Adult cats (1.5–3.5 kg) were initially anesthetized with a dose of propofol and sufentanil (20 mg/kg + 1.5 µg/kg, i.v.) that was reduced to (5 + 1.5 µg/kg/h, i.v.) for maintenance. The depth of anesthesia was monitored by electrocardiogram, EEG, and absence of an autonomic response; body temperature was held near 37 °C. After surgical procedures, the animal was paralyzed with vecuronium bromide (0.2 mg/kg/h, i.v.) and ventilated artificially; expired $CO_2$ and blood oxygenation were monitored throughout the experiment. Pupils were dilated with 1% atropine sulfate and the nictitating membranes retracted with 10% phenylephrine. The eyes were refracted and fitted with contact lenses to focus on a tangent screen on which the positions of the area centralis and the optic disk of each eye were marked. A craniotomy centered on Horsley–Clark coordinates A6.5 and L8.5 gave access to the visual thalamus. All procedures were in accordance with the guidelines of the National Institute of Health and the Institutional Animal Care and Use Committee of the University of Southern California.

**Multi-electrode recordings**. We first used a tungsten electrode to locate the LGN to guide placement of a multi-electrode array[49]. The array was made of seven independently movable quartz–platinum electrodes (3–4 MΩ impedance, 80 µm diameter) arranged in either a concentric or linear configuration, spaced 305 µm apart. These were lowered into the brain through a metal guide tube (tip diameter 1.1 mm, length 11 mm) whose tip rested ~5 mm above the LGN, and, hence, several mm above the PGN[28]. The angle of the multi-electrode was adjusted (25–30° anterior–posterior and 5° medial–lateral) so that the electrode travelled through the same retinotopic position in the thalamus (Fig. 1a)[50]. Signals from all electrodes were amplified, filtered, and stored on a computer running

Spike2 software (Cambridge Electronic Design, Cambridge, UK), which we used to extract action potentials for subsequent analysis.

**Identification of cells in the PGN and LGN**. We followed standard in vivo procedure of identification of reticular units as published before[21], utilizing a combination of physiological and anatomical criteria, viz. binocularity, overlapping on and off responses[3, 7, 18–20, 28], structure of burst firing (Fig. 1d),[12, 13, 21] and post hoc identification of LGN typically < 1 mm deeper than the recording site. In addition, we made confirmed recording sites by locating the electrode tracks and/or electrolytic lesions (Fig. 1a, *inset*) in Nissl-stained sections of the PGN.

**Visual stimulus**. The visual stimulus was displayed on a monochrome cathode ray tube monitor (refresh rate 144 Hz) 915 mm from the eyes, and controlled by the ViSaGe stimulus generator (Cambridge Research Design, Ltd., Cambridge, UK). The stimulus was spatiotemporal Gaussian white noise, with a frame rate of 48 Hz. The intensity of each stixel (spatiotemporal pixel) in the stimulus grid was sampled from a Gaussian distribution (at 33% root-mean-square contrast), and stixel size was 0.5° or 1°. The stimulus sequence was 16 384 frames long.

**Estimation of the spatial scale of receptive fields**. The shapes of receptive fields in the PGN were amorphous and diverse, so it was not possible to use standard parametric fitting functions (e.g., a two-dimensional Gaussian) to quantify their sizes. In order to develop a general approach to quantifying receptive field size that could be applied to both the LGN and PGN, we modified statistical methods used to analyze data from fMRI[29]. First, we used standard methods of reverse correlation to obtain the spatio–temporal receptive field by computing the STA of the stimulus ensemble[30]. We then assessed the statistical significance of each stixel (i.e., a voxel in three-dimensional space–time) in the spatio–temporal receptive field with a bootstrap resampling approach[30]. Next, we computed the $Z$-scores and corresponding $q$-values for each stixel (Fig. 3d). Then, we identified significant on and off stixels using the false discovery rate test[51] on all the $q$-values, at a significance level of 0.01 (Fig. 2e). Finally, we collapsed the temporal dimension to yield a spatial map of significant pixels, which we called the "footprint" of a receptive field (Fig. 2f). Thus, the spatial size of the on, off or combined on–off receptive fields could be quantified by the area covered by the corresponding "footprint" (Fig. 3f). All analyses were done with Matlab (Mathworks, Natick, MA).

**Data availability**. Our data regarding receptive field sizes are available on request.

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

## Acknowledgements

The work was supported by NIH EY09593 to J.A.H., NSF 1219212 to F.T.S., a Life Sciences Research Foundation's Pfizer Fellowship to X.W., and a MEC/Fulbright to C.S.-S.

## Author contributions

C.S.-S., V.V., and J.A.H. performed the experiments. C.S.-S. analyzed the data, and X.W. and F.T.S. designed new analytical tools. All authors contributed to the experimental design, and C.S.-S., X.W., and J.A.H. wrote the manuscript.

## Additional information

**Competing interests:** The authors declare no competing financial interests.

