## [Peer Review File · Nature Communications]

Reviewers' Comments:

Reviewer #1 (Remarks to the Author):

The manuscript by Soto-Sanchez provides a quantitative description for the visual properties of the perigeniculate nucleus (PGN) neurons of the cat. Using an extremely challenging dataset to obtain, they report that PGN neurons do not have amorphous and ill-defined receptive fields as might have been previously suspected, but rather well-defined ones whose spatial scale is roughly equivalent to those of neighboring LGN neurons.

This manuscript deserves rapid and broad dissemination for the following two reasons:

1. It provides a non-incremental advance in our knowledge of thalamic inhibition at a single cell resolution. Studying thalamic inhibition has gained substantial momentum with the explosion of molecular tools available to interrogate the reticular thalamus and thalamic interneurons. However, many of these studies are limited by the mouse as the model system, and the current study provides a unique advantage in investigating a highly visual animal (the cat), where interpretation of spatial scale is likely to be much more closely related to primates (including us). This dataset will therefore be invaluable in future comparative studies across both rodents and primates.
2. The methods used to extract the receptive field properties are done at the highest standards in the field, and for whatever reason, have not been adopted widely in studying thalamic inhibition. Meaning, previous studies did not use reverse correlation in order to determine quantitative properties of PGN (visual sector of TRN) sensory responses, and therefore had contributed quite a fair bit of confusion to guiding our thinking how the TRN contributes to sensory processing. This paper provides a great deal of clarity that will undoubtedly propel our thinking moving forward.

Given my enthusiasm for this work, I would like to provide the authors with feedback that I think would help with their manuscript receiving the attention it deserves. There is some additional analysis that would be helpful, and I think that some reframing of certain ideas is necessary.

Reframing:

This manuscript does not settle the score on thermostat vs. searchlight. It is not any less

important from what I articulated above for not having done so. Instead, I think it provides an excellent start along the way to resolving this debate. The authors, I am sure, are keenly aware that to understand thermostat vs. searchlight, we would also need to know what the output of TRN, not only its inputs (as assessed by the STA). This is of course, all in the context of pure sensory processing, and how top-down inputs may change the type of TRN engagement in sensory processing is a whole other topic. As for the output, things like how many LGN neurons does a single TRN cell target? How does it exactly modify each one of these neuron's sensory responses? How do they change with varying behavioral states and demands? Clearly these questions are beyond the scope of this manuscript, but it should be made clear somewhere (maybe the discussion?), that these are required to fully settle that debate, in addition to the very nice results presented in this paper.

Analysis:

The footprint analysis is basically z-scoring of the STA at $p < 0.01$ (Z-score of 2.58). The spatial scale comparison between LGN and PGN footprints is the sum of that footprint. I am not sure I fully understand the rationale for this, because the authors may be missing some key insights:

1. Why $p < 0.01$? The On and Off responses are, by definition, one sided-- so the comparison really shouldn't be at a p intended for a two-sided comparison. Also, it is quite conservative. What if the authors were to simply lower that threshold. I can tell, based on the multiple nice examples in this manuscript that the PGN neurons are much more multi-peaked than LGN neurons. Now, individual peaks themselves are unlikely to be larger than the individual peaks in LGN based on visual inspection, but there would be many more of them in PGN than LGN. At the very least, the authors would be well advised to do their z-score thresholding at multiple values, because that could be helpful for a broader interpretation of the data in future studies.

2. Related to my point above, even with the current threshold, the authors only report footprint size. I bet that if they were to measure the number of peaks (based on some minimum number of spatially contiguous pixels) that PGN neurons would win over LGN. This is useful information that is currently neglected. At the very least, it may tell us that PGN neurons derive their visual responses from many more presynaptic inputs (that also seem to be spatially diverse), than the more spatially-contiguous retinal inputs LGN neurons receive. In fact, it may even be that individual peak sizes of PGN neurons end up being smaller than LGN, which is also important information.

Summary: This is an exciting and important addition to the fields of vision, thalamus and inhibitory control. I am fully supportive of rapid and broad dissemination, and would hope that the authors find my feedback helpful.

Reviewer #2 (Remarks to the Author):

Soto-Sánchez et al have studied the visual receptive field of perigeniculate neurons in anesthetized cats. Using well-isolated single unit recordings that unambiguously identify individual neurons they test whether the receptive fields of PGN neurons might be broadly or narrowly tuned in terms of visual space. The latter type of organization was proposed as one of two possibilities by Crick. Crick also hypothesized that PGN (and the related TRN) may serve as a searchlight, which would require spatially restricted receptive fields. The authors adapt a statistical approach to test receptive field size, and find that indeed the latter hypothesis is supported. Overall the work is done well, as expected for this research group.

I have only two comments regarding the current ms, in which the data provided do support the conclusions.

1) The issue of anesthesia obviously needs to be addressed, at least in the discussion. The important issue can no longer be ignored by investigators in systems neuroscience. It is abundantly clear that receptive field properties are dramatically influenced by arousal level and anesthetic state.

2) The authors appear to be unaware of work on RTN receptive fields outside of the visual system. This oversight should be corrected. E.g. PMID 10805677.

Reviewer #3 (Remarks to the Author):

The study by Soto-Sanchez and colleagues on “Visual neurons in the cat’s thalamic reticular nucleus operate over local spatial scales” used reversed correlation methods to measure detailed spatial receptive fields in the cat’s lateral geniculate nucleus (LGN) relative to the perigeniculate neurons. The main finding is that neurons of the cat’s thalamic reticular nucleus (TRN) have small receptive fields that match in terms of extent their counterparts in the LGN. While this study has been thoroughly performed and uses state-of-the-art methods, the result is not particularly novel or unexpected. In addition, I have a number of concerns regarding the framing of the study, as outlined more specifically below.

1) The study sets out to test several competing hypothesis about the functional role of the TRN, namely a role to operate as a ‘thermostat’ as compared to a ‘searchlight’. While both of these metaphors offer reasonable underlying mechanistic constructs, I was not clear why they are mutually exclusive. The TRN receives a great variety of cortical inputs from many different

brain regions and networks and may very well be able to switch such functional roles depending on behavioral demands by recruiting different input to TRN. The concept that the visual TRN has 'one function' is overtly simplistic.

2) Much weight is given to the fact that local receptive field (RF) structure supports a searchlight function. For several reasons, I am not convinced that this is necessarily true. First and foremost, the case should be made in an awake and behaving animal and not in an anesthetized preparation. RFs could change dramatically depending on behavioral demands (see literature in non-human primates on the topic). Thus, the RF sizes in an anesthetized animal do not tell us anything about how they might change in a behaving state. This issue is a major limitation of the study. Second, neurons in primate frontal and parietal cortex are part of an extensive attention network that operates as a searchlight despite the fact that their RFs are extensive and not confined to local scales.

3) The number of neurons tested is relatively small (<100) and from a large number of animals. Thus, the sample assumes that eccentricity (magnification) functions are identical across individuals. I am skeptical about this assumption. In human neuroimaging studies, there is a wide individual variability. Thus, the sampling approach (i.e. few neurons per animal) may have introduced noise with respect to the eccentricity functions.

Response to reviewers

Reviewer #1 (Remarks to the Author):

perigeniculate nucleus (PGN) neurons of the cat. Using an extremely challenging dataset to obtain, they report that PGN neurons do not have amorphous and ill-defined receptive fields as might have been previously suspected, but rather well-defined ones whose spatial scale is roughly equivalent to those of neighboring LGN neurons.

This manuscript deserves rapid and broad dissemination for the following two reasons:

1. It provides a non-incremental advance in our knowledge of thalamic inhibition at a single cell resolution. Studying thalamic inhibition has gained substantial momentum with the explosion of molecular tools available to interrogate the reticular thalamus and thalamic interneurons. However, many of these studies are limited by the mouse as the model system, and the current study provides a unique advantage in investigating a highly visual animal (the cat), where interpretation of spatial scale is likely to be much more closely related to primates (including us). This dataset will therefore be invaluable in future comparative studies across both rodents and primates.

2. The methods used to extract the receptive field properties are done at the highest standards in the field, and for whatever reason, have not been adopted widely in studying thalamic inhibition. Meaning, previous studies did not use reverse correlation in order to determine quantitative properties of PGN (visual sector of TRN) sensory responses, and therefore had contributed quite a fair bit of confusion to guiding our thinking how the TRN contributes to sensory processing. This paper provides a great deal of clarity that will undoubtedly propel our thinking moving forward.

Given my enthusiasm for this work, I would like to provide the authors with feedback that I think would help with their manuscript receiving the attention it deserves. There is some additional analysis that would be helpful, and I think that some reframing of certain ideas is necessary.

We thank the Reviewer for the recognition of the importance of our work. We revised the manuscript per the Reviewer's suggestions, as below.

Reframing:

This manuscript does not settle the score on thermostat vs. searchlight. It is not any less important from what I articulated above for not having done so. Instead, I think it provides an excellent start along the way to resolving this debate. The authors, I am sure, are keenly aware that to understand thermostat vs. searchlight, we would also need to know what the output of TRN, not only its inputs (as assessed by the STA). This is of course, all in the context of pure sensory processing, and how top-down inputs may

change the type of TRN engagement in sensory processing is a whole other topic. As for the output, things like how many LGN neurons does a single TRN cell target? How does it exactly modify each one of these neuron's sensory responses? How do they change with varying behavioral states and demands? Clearly these questions are beyond the scope of this manuscript, but it should be made clear somewhere (maybe the discussion?), that these are required to fully settle that debate, in addition to the very nice results presented in this paper.

We thank the Reviewer for these suggestions. While we still discuss the work in the context of Crick's competing hypotheses, we explain that this framework provides useful a starting point for our study, but is not the endpoint. Moreover, we are careful to emphasize that a key prediction of the searchlight hypothesis—that reticular receptive fields are spatially localized—has equally important implications for sensory processing *per se*.

We added a new paragraph (excerpted below) to the Discussion to address questions that the reviewer noted must be answered to further understanding of the role of the TRN in attention and processing.

...“Of course, a fuller test of the roles of the PGN awaits further experiments. It will be important to identify the inputs and outputs of each reticular cell and then determine how functional connectivity changes as a function of stimulus context, behavioral state and task demands.”

Analysis:

The footprint analysis is basically z-scoring of the STA at $p < 0.01$ (Z-score of 2.58). The spatial scale comparison between LGN and PGN footprints is the sum of that footprint. I am not sure I fully understand the rationale for this, because the authors may be missing some key insights:

We thank the Reviewer for the thoughtful comments, which we address below.

1. Why $p < 0.01$? The On and Off responses are, by definition, one sided-- so the comparison really shouldn't be at a p intended for a two-sided comparison. Also, it is quite conservative. What if the authors were to simply lower that threshold. I can tell, based on the multiple nice examples in this manuscript that the PGN neurons are much more multi-peaked than LGN neurons. Now, individual peaks themselves are unlikely to be larger than the individual peaks in LGN based on visual inspection, but there would be many more of them in PGN than LGN. At the very least, the authors would be well advised to do their z-score thresholding at multiple values, because that could be helpful for a broader interpretation of the data in future studies.

For our analysis, it seems as if we conveyed the message that, for each cell, we determined whether the preferred polarity was On or Off, and then inverted the sign of the Off maps so the Z scores we calculated would all have the same, positive, sign. Had we done so, we could have used a one-sided test. However, we simply implemented our analysis agnostically for On and Off cells. Thus the resulting distribution included positive and negative Z-scores (according to the cell's polarity) and we needed to use a two-sided test to identify significant Z values.

We used Benjamini-Hochberg control of false discover rate (FDR) at a level of $q = 0.01$, which, as the Reviewer noted, was an arbitrary threshold. Per the Reviewer's request, we repeated the analysis with a more stringent ($q = 0.001$) as well as a more relaxed value ($q = 0.05$) threshold. The size of the footprints changed slightly, but the relationship between the relative sizes of footprints in the LGN vs PGN remained the same. We have included these additional analyses in Supplementary Fig. 2. We probably (and inadvertently) confused the Reviewer by substituting "p" for "q" because we thought this might be clearer for the general reader. We have revised the text and used "q".

2. Related to my point above, even with the current threshold, the authors only report footprint size. I bet that if they were to measure the number of peaks (based on some minimum number of spatially contiguous pixels) that PGN neurons would win over LGN. This is useful information that is currently neglected. At the very least, it may tell us that PGN neurons derive their visual responses from many more presynaptic inputs (that also seem to be spatially diverse), than the more spatially-contiguous retinal inputs LGN neurons receive. In fact, it may even be that individual peak sizes of PGN neurons end up being smaller than LGN, which is also important information.

We have quantified the number of spatially contiguous regions within the receptive fields and created a new figure for the supplementary materials (Supplementary Fig. 2). For LGN, the number of contiguous patches in the footprint region representing the preferred stimulus contrast (the center, in effect) is usually near one, for $q = 0.01$. The number of patches for a stimulus of the non-dominant polarity is usually greater than one, consistent with the often patchy appearance of the surround. For PGN, there are multiple contiguous patches for both contrasts. This finding is consistent with the idea that reticular cells pool input from many relay cells. We are hesitant to suggest that each patch or peak in the reticular footprint corresponds to input from a single relay cell, however, the data are there for the reader to consider.

Summary: This is an exciting and important addition to the fields of vision, thalamus and inhibitory control. I am fully supportive of rapid and broad dissemination, and would hope that the authors find my feedback helpful.

Reviewer #2 (Remarks to the Author):

Soto-Sánchez et al have studied the visual receptive field of perigeniculate neurons in anesthetized cats. Using well-isolated single unit recordings that unambiguously identify individual neurons they test whether the receptive fields of PGN neurons might be broadly or narrowly tuned in terms of visual space. The latter type of organization was proposed as one of two possibilities by Crick. Crick also hypothesized that PGN (and the related TRN) may serve as a searchlight, which would require spatially restricted receptive fields. The authors adapt a statistical approach to test receptive field size, and find that indeed the latter hypothesis is supported. Overall the work is done well, as expected for this research group.

I have only two comments regarding the current ms, in which the data provided do support the conclusions.

1) The issue of anesthesia obviously needs to be addressed, at least in the discussion. The important issue can no longer be ignored by investigators in systems neuroscience. It is abundantly clear that receptive field properties are dramatically influenced by arousal level and anesthetic state.

Thank you. The revised discussion addresses how anesthesia, the degree of synchronization of the EEG and behavioral state can influence receptive field size. In particular we cite a study that compares spatial frequency preference in the anesthetized vs awake monkey (spatial frequency tuning shifts towards higher values, at least in the magno pathway, for awake animals) ¹ and another that shows that cortical receptive field size grows smaller as the EEG desynchronizes ². Thus, one would expect that the PGN receptive fields we measured in the anesthetized animal might be larger than in the awake state. This is consistent with the conclusion that reticular neurons operate over local spatial scales.

2) The authors appear to be unaware of work on RTN receptive fields outside of the visual system. This oversight should be corrected. E.g. PMID 10805677.

We are, of course, aware of work in the reticular nucleus outside of vision and had addressed the auditory and somatosensory sectors of the TRN in the first submission and in past work. In the revision, we discuss the other modalities more explicitly and have included the Hartings citation (previously referenced only indirectly in a review article ³). We have excerpted relevant parts of the manuscript below.

“Spike-triggered average and covariance analyses have shown that neurons in the PGN are, in fact, highly selective and are tuned for specific features²², as are cells in the auditory³² and somatosensory divisions of the TRN³⁸ “

“Our study involved only the visual sector of the TRN. Although the spatial scale of receptive fields in auditory or somatosensory TRN have not been measured, reticular cells there are tuned for specific features^{38, 49}. These findings suggest that strategies for processing in the visual TRN are conserved across sensory modalities.”

Reviewer #3 (Remarks to the Author):

The study by Soto-Sanchez and colleagues on “Visual neurons in the cat’s thalamic reticular nucleus operate over local spatial scales” used reversed correlation methods to measure detailed spatial receptive fields in the cat’s lateral geniculate nucleus (LGN) relative to the perigeniculate neurons. The main finding is that neurons of the cat’s thalamic reticular nucleus (TRN) have small receptive fields that match in terms of extent their counterparts in the LGN. While this study has been thoroughly performed and uses state-of-the-art methods, the result is not particularly novel or unexpected. In addition, I have a number of concerns regarding the framing of the study, as outlined more specifically below.

We do not understand how the reviewer could guess *a priori* that our results reveal localized receptive fields in the TRN. The spatial scale of receptive fields in the TRN in general, and PGN, in particular, has never been quantified and many previous investigators had assumed that their size was far larger than that of relay cells e.g^{4,5}.

1) The study sets out to test several competing hypothesis about the functional role of the TRN, namely a role to operate as a ‘thermostat’ as compared to a ‘searchlight’. While both of these metaphors offer reasonable underlying mechanistic constructs, I was not clear why they are mutually exclusive. The TRN receives a great variety of cortical inputs from many different brain regions and networks and may very well be able to switch such functional roles depending on behavioral demands by recruiting different input to TRN. The concept that the visual TRN has ‘one function’ is overtly simplistic.

We agree that the two competing hypotheses, as originally formulated by Crick, are not mutually exclusive in all regards. But the two hypotheses make very different predictions about the size of reticular receptive fields. The spotlight hypothesis necessitates feedback that conveys local information. The global thermostat hypothesis requires feedback that conveys spatially distributed information. Global information could be conveyed either by cells with large

receptive fields or by a high degree of convergence. However, the anatomical observation that the projections fed back from PGN to LGN are topographically compact argues against the latter possibility ⁴. These points are strengthened in the revision, see Discussion. We provide a sample excerpt below:

“Ultimately, our systematic analyses establish that reticular receptive fields are roughly the same size as those of relay cells at a given eccentricity. This finding is in keeping with the searchlight hypothesis and suggests that if inhibitory feedback from the PGN extends far beyond the relay cell’s classical receptive field, it is likely mediated by the pooled input of cells tuned to complex features²².”

We never suggested that TRN has only one function; our original submission noted varied potential roles. In the revision, we explain that our current finding of small receptive fields in TRN, in addition to past work showing that neurons in the TRN are sensitive to specific stimulus features ⁶, suggests an important role in feature selective sensory processing *per se*. We also now note a potential role in bottom-up processing ⁷. Comments regarding role for the TRN are included throughout the revised manuscript.

2) Much weight is given to the fact that local receptive field (RF) structure supports a searchlight function. For several reasons, I am not convinced that this is necessarily true. First and foremost, the case should be made in an awake and behaving animal and not in an anesthetized preparation. RFs could change dramatically depending on behavioral demands (see literature in non-human primates on the topic). Thus, the RF sizes in an anesthetized animal do not tell us anything about how they might change in a behaving state. This issue is a major limitation of the study.

Please see our response to Reviewer 2 regarding literature that shows that spatial extent of receptive fields in the LGN and V1 shrinks as the animal becomes more alert.

Second, neurons in primate frontal and parietal cortex are part of an extensive attention network that operates as a searchlight despite the fact that their RFs are extensive and not confined to local scales.

Although top-down attentional modulation of feedforward sensory information processing involves many stations in the brain, there is substantial evidence that modulation of spatial attention in the TRN operates over local spatial scales ⁸.

3) The number of neurons tested is relatively small (<100) and from a large number of animals. Thus, the sample assumes that eccentricity (magnification) functions are identical across individuals. I am skeptical about this assumption. In human neuroimaging studies, there is a wide individual variability. Thus, the sampling approach (i.e. few neurons per animal) may have introduced noise with respect to the eccentricity functions.

We measured the eccentricity of the receptive fields from the precise location of the *area centralis* projected on a tangent screen. That is, the measurements of eccentricity were made by using retinal coordinates, rather than by using the position of the electrodes in the brain. Thus, the magnification factor did not introduce error into our measurements.

In any event, to address the Reviewer's request, we combed through our database and found additional cells that we had recorded. Our sample is now 88 cells. This is a substantial number of recordings, especially given the difficulty of recording from a slim and cell-sparse structure. We have updated the figures to reflect this increased dataset.

References

1. Alitto, H.J., Moore, B.D., Rathbun, D.L. & Usrey, W.M. A comparison of visual responses in the lateral geniculate nucleus of alert and anaesthetized macaque monkeys. *The Journal of Physiology* **589**, 87-99 (2011).
2. Worgotter, F., *et al.* State-dependent receptive-field restructuring in the visual cortex. *Nature* **396**, 165-168 (1998).
3. Pinault, D. The thalamic reticular nucleus: structure, function and concept. *Brain Research Reviews* **46**, 1-31 (2004).
4. Uhlich, D.J., Cucchiaro, J.B., Humphrey, A.L. & Sherman, S.M. Morphology and axonal projection patterns of individual neurons in the cat perigeniculate nucleus. *J. Neurophysiol.* **65**, 1528-1541 (1991).
5. Bonin, V., Mante, V. & Carandini, M. The suppressive field of neurons in lateral geniculate nucleus. *J. Neurosci.* **25**, 10844-10856 (2005).
6. Vaingankar, V., Soto Sanchez, C., Wang, X., Sommer, F.T. & Hirsch, J.A. Neurons in the thalamic reticular nucleus are selective for diverse and complex visual features. *Front. Integr. Neurosci.* **6** (2012).
7. Frey, H.P., Konig, P. & Einhauser, W. The role of first- and second-order stimulus features for human overt attention. *Perception and Psychophysics* **69**, 153-161 (2007).
8. McAlonan, K., Cavanaugh, J. & Wurtz, R.H. Guarding the gateway to cortex with attention in visual thalamus. *Nature* (2008).

Reviewers' Comments:

Reviewer #1 (Remarks to the Author):

The authors have addressed all my comments. Congrats on a really cool piece of work!

Reviewer #2 (Remarks to the Author):

The authors have fully responded to previous concerns. This is an important piece of work and merits timely publication.

Reviewer #3 (Remarks to the Author):

The authors have made a laudable attempt to strengthen the study. While several of my original points have not been fully addressed, because they need a different level of experimentation (e.g. awake preparation), I do think that the current study is an important advance and deserves rapid publication.